# Reparameterization Gradient
# for Non-differentiable Models

**Wonyeol Lee**   **Hangyeol Yu**   **Hongseok Yang**
School of Computing, KAIST
Daejeon, South Korea
{wonyeol, yhk1344, hongseok.yang}@kaist.ac.kr

## Abstract

We present a new algorithm for stochastic variational inference that targets at models with non-differentiable densities. One of the key challenges in stochastic variational inference is to come up with a low-variance estimator of the gradient of a variational objective. We tackle the challenge by generalizing the reparameterization trick, one of the most effective techniques for addressing the variance issue for differentiable models, so that the trick works for non-differentiable models as well. Our algorithm splits the space of latent variables into regions where the density of the variables is differentiable, and their boundaries where the density may fail to be differentiable. For each differentiable region, the algorithm applies the standard reparameterization trick and estimates the gradient restricted to the region. For each potentially non-differentiable boundary, it uses a form of manifold sampling and computes the direction for variational parameters that, if followed, would increase the boundary's contribution to the variational objective. The sum of all the estimates becomes the gradient estimate of our algorithm. Our estimator enjoys the reduced variance of the reparameterization gradient while remaining unbiased even for non-differentiable models. The experiments with our preliminary implementation confirm the benefit of reduced variance and unbiasedness.

## 1   Introduction

Stochastic variational inference (SVI) is a popular choice for performing posterior inference in Bayesian machine learning. It picks a family of variational distributions, and formulates posterior inference as a problem of finding a member of this family that is closest to the target posterior. SVI, then, solves this optimization problem approximately using stochastic gradient ascent. One major challenge in developing an effective SVI algorithm is the difficulty of designing a low-variance estimator for the gradient of the optimization objective. Addressing this challenge has been the driver of recent advances for SVI, such as reparameterization trick [13, 30, 31, 26, 15], clever control variate [28, 7, 8, 34, 6, 23], and continuous relaxation of discrete distributions [20, 10].

Our goal is to tackle the challenge for models with non-differentiable densities. Such a model naturally arises when one starts to use both discrete and continuous random variables or specifies a model using programming constructs, such as if statement, as in probabilistic programming [4, 22, 37, 5]. The high variance of a gradient estimate is a more serious issue for these models than for those with differentiable densities. Key techniques for addressing it simply do not apply in the absence of differentiability. For instance, a prerequisite for the so called reparameterization trick is the differentiability of a model's density function.

In the paper, we present a new gradient estimator for non-differentiable models. Our estimator splits the space of latent variables into regions where the joint density of the variables is differentiable, and their boundaries where the density may fail to be differentiable. For each differentiable region, the estimator applies the standard reparameterization trick and estimates the gradient restricted to the

region. For each potentially non-differentiable boundary, it uses a form of manifold sampling, and computes the direction for variational parameters that, if followed, would increase the boundary's contribution to the variational objective. This manifold sampling step cannot be skipped if we want to get an unbiased estimator, and it only adds a linear overhead to the overall estimation time for a large class of non-differentiable models. The result of our gradient estimator is the sum of all the estimated values for regions and boundaries.

Our estimator generalizes the estimator based on the reparameterization trick. When a model has a differentiable density, these two estimators coincide. But even when a model's density is not differentiable and so the reparameterization estimator is not applicable, ours still applies; it continues to be an unbiased estimator, and enjoys variance reduction from reparameterization. The unbiasedness of our estimator is not trivial, and follows from an existing yet less well-known theorem on exchanging integration and differentiation under moving domain [3] and the divergence theorem. We have implemented a prototype of an SVI algorithm that uses our gradient estimator and works for models written in a simple first-order loop-free probabilistic programming language. The experiments with this prototype confirm the strength of our estimator in terms of variance reduction.

## 2   Variational Inference and Reparameterization Gradient

Before presenting our results, we review the basics of stochastic variational inference.

Let $\boldsymbol{x}$ and $\boldsymbol{z}$ be, respectively, observed and latent variables living in $\mathbb{R}^m$ and $\mathbb{R}^n$, and $p(\boldsymbol{x}, \boldsymbol{z})$ a density that specifies a probabilistic model about $\boldsymbol{x}$ and $\boldsymbol{z}$. We are interested in inferring information about the posterior density $p(\boldsymbol{z}|\boldsymbol{x}^0)$ for a given value $\boldsymbol{x}^0$ of $\boldsymbol{x}$.

Variational inference approaches this posterior-inference problem from the optimization angle. It recasts posterior inference as a problem of finding a best approximation to the posterior among a collection of pre-selected distributions $\{q_\theta(\boldsymbol{z})\}_{\theta \in \mathbb{R}^d}$, called *variational distributions*, which all have easy-to-compute and easy-to-differentiate densities and permit efficient sampling. A standard objective for this optimization is to maximize a lower bound of $\log p(\boldsymbol{x}^0)$ called *evidence lower bound* or simply ELBO:

$$\mathrm{argmax}_\theta\Big(\mathsf{ELBO}_\theta\Big), \quad \text{where } \mathsf{ELBO}_\theta \triangleq \mathbb{E}_{q_\theta(\boldsymbol{z})}\left[\log \frac{p(\boldsymbol{x}^0, \boldsymbol{z})}{q_\theta(\boldsymbol{z})}\right]. \tag{1}$$

It is equivalent to the objective of minimizing the KL divergence from $q_\theta(\boldsymbol{z})$ to the posterior $p(\boldsymbol{z}|\boldsymbol{x}^0)$.

Most of recent variational-inference algorithms solve the optimization problem (1) by stochastic gradient ascent. They repeatedly estimate the gradient of $\mathsf{ELBO}_\theta$ and move $\theta$ towards the direction of this estimate:

$$\theta \leftarrow \theta + \eta \cdot \widehat{\nabla_\theta \mathsf{ELBO}_\theta}$$

The success of this iterative scheme crucially depends on whether it can estimate the gradient well in terms of computation time and variance. As a result, a large part of research efforts on stochastic variational inference has been devoted to constructing low-variance gradient estimators or reducing the variance of existing estimators.

The reparameterization trick [13, 30] is the technique of choice for constructing a low-variance gradient estimator for models with differentiable densities. It can be applied in our case if the joint $p(\boldsymbol{x}, \boldsymbol{z})$ is differentiable with respect to the latent variable $\boldsymbol{z}$. The trick is a two-step recipe for building a gradient estimator. First, it tells us to find a distribution $q(\boldsymbol{\epsilon})$ on $\mathbb{R}^n$ and a smooth function $f : \mathbb{R}^d \times \mathbb{R}^n \to \mathbb{R}^n$ such that $f_\theta(\boldsymbol{\epsilon})$ for $\boldsymbol{\epsilon} \sim q(\boldsymbol{\epsilon})$ has the distribution $q_\theta$. Next, the reparameterization trick suggests us to use the following estimator:

$$\widehat{\nabla_\theta \mathsf{ELBO}_\theta} \triangleq \frac{1}{N}\sum_{i=1}^{N} \nabla_\theta \log \frac{r(f_\theta(\boldsymbol{\epsilon}^i))}{q_\theta(f_\theta(\boldsymbol{\epsilon}^i))}, \quad \text{where } r(\boldsymbol{z}) \triangleq p(\boldsymbol{x}^0, \boldsymbol{z}) \text{ and } \boldsymbol{\epsilon}^1, \dots, \boldsymbol{\epsilon}^N \sim q(\boldsymbol{\epsilon}). \tag{2}$$

The reparameterization gradient in (2) is unbiased, and has variance significantly lower than the so called score estimator (or REINFORCE) [35, 27, 36, 28], which does not exploit differentiability. But so far its use has been limited to differentiable models. We will next explain how to lift this limitation.

# 3   Reparameterization for Non-differentiable Models

Our main result is a new unbiased gradient estimator for a class of non-differentiable models, which can use the reparameterization trick despite the non-differentiability.

Recall the notations from the previous section: $\boldsymbol{x} \in \mathbb{R}^m$ and $\boldsymbol{z} \in \mathbb{R}^n$ for observed and latent variables, $p(\boldsymbol{x}, \boldsymbol{z})$ for their joint density, $\boldsymbol{x}^0$ for an observed value, and $q_\theta(\boldsymbol{z})$ for a variational distribution parameterized by $\theta \in \mathbb{R}^d$.

Our result makes two assumptions. First, the variational distribution $q_\theta(\boldsymbol{z})$ satisfies the conditions of the reparameterization gradient. Namely, $q_\theta(\boldsymbol{z})$ is continuously differentiable with respect to $\theta \in \mathbb{R}^d$, and is the distribution of $f_\theta(\boldsymbol{\epsilon})$ for a smooth function $f : \mathbb{R}^d \times \mathbb{R}^n \to \mathbb{R}^n$ and a random variable $\boldsymbol{\epsilon} \in \mathbb{R}^n$ distributed by $q(\boldsymbol{\epsilon})$. Also, the function $f_\theta$ on $\mathbb{R}^n$ is bijective for every $\theta \in \mathbb{R}^d$. Second, the joint density $r(\boldsymbol{z}) = p(\boldsymbol{x}^0, \boldsymbol{z})$ at $\boldsymbol{x} = \boldsymbol{x}^0$ has the following form:

$$r(\boldsymbol{z}) = \sum_{k=1}^{K} \mathbb{1}[\boldsymbol{z} \in R_k] \cdot r_k(\boldsymbol{z}) \tag{3}$$

where $r_k$ is a non-negative continuously-differentiable function $\mathbb{R}^n \to \mathbb{R}$, $R_k$ is a (measurable) subset of $\mathbb{R}^n$ with measurable boundary $\partial R_k$ such that $\int_{\partial R_k} d\boldsymbol{z} = 0$, and $\{R_k\}_{1 \le k \le K}$ is a partition of $\mathbb{R}^n$. Note that $r(\boldsymbol{z})$ is an unnormalized posterior under the observation $\boldsymbol{x} = \boldsymbol{x}^0$. The assumption indicates that the posterior $r$ may be non-differentiable at some $\boldsymbol{z}$'s, but all the non-differentiabilities occur only at the boundaries $\partial R_k$ of regions $R_k$. Also, it ensures that when considered under the usual Lebesgue measure on $\mathbb{R}^n$, these non-differentiable points are negligible (i.e., they are included in a null set of the measure). As we illustrate in our experiments section, models satisfying our assumption naturally arise when one starts to use both discrete and continuous random variables or specifies models using programming constructs, such as if statement, as in probabilistic programming [4, 22, 37, 5].

Our estimator is derived from the following theorem:

**Theorem 1.** *Let*

$$h_k(\boldsymbol{\epsilon}, \theta) \triangleq \log \frac{r_k(f_\theta(\boldsymbol{\epsilon}))}{q_\theta(f_\theta(\boldsymbol{\epsilon}))}, \qquad \boldsymbol{V}(\boldsymbol{\epsilon}, \theta) \in \mathbb{R}^{d \times n}, \qquad \boldsymbol{V}(\boldsymbol{\epsilon}, \theta)_{ij} \triangleq \left( \frac{\partial}{\partial \theta_i} \left( f_\theta^{-1}(\boldsymbol{z}) \Big|_{\boldsymbol{z} = f_\theta(\boldsymbol{\epsilon})} \right) \right)_j.$$

*Then,*

$$\nabla_\theta \mathsf{ELBO}_\theta = \underbrace{\mathbb{E}_{q(\boldsymbol{\epsilon})} \left[ \sum_{k=1}^{K} \mathbb{1}[f_\theta(\boldsymbol{\epsilon}) \in R_k] \cdot \nabla_\theta h_k(\boldsymbol{\epsilon}, \theta) \right]}_{\mathsf{RepGrad}_\theta} + \underbrace{\sum_{k=1}^{K} \int_{f_\theta^{-1}(\partial R_k)} \left( q(\boldsymbol{\epsilon}) h_k(\boldsymbol{\epsilon}, \theta) \boldsymbol{V}(\boldsymbol{\epsilon}, \theta) \right) \bullet d\boldsymbol{\Sigma}}_{\mathsf{BouContr}_\theta}$$

*where the RHS of the plus uses the surface integral of $q(\boldsymbol{\epsilon}) h_k(\boldsymbol{\epsilon}, \theta) \boldsymbol{V}(\boldsymbol{\epsilon}, \theta)$ over the boundary $f_\theta^{-1}(\partial R_k)$ expressed in terms of $\boldsymbol{\epsilon}$, the $d\boldsymbol{\Sigma}$ is the normal vector of this boundary that is outward pointing with respect to $f_\theta^{-1}(R_k)$, and the $\bullet$ operation denotes the matrix-vector multiplication.*

The theorem says that the gradient of $\mathsf{ELBO}_\theta$ comes from two sources. The first is the usual reparameterized gradient of each $h_k$ but restricted to its region $R_k$. The second source is the sum of the surface integrals over the region boundaries $\partial R_k$. Intuitively, the surface integral for $k$ computes the direction to move $\theta$ in order to increase the contribution of the boundary $\partial R_k$ to $\mathsf{ELBO}_\theta$. Note that the integrand of the surface integral has the additional $\boldsymbol{V}$ term. This term is a by-product of rephrasing the original integration over $\boldsymbol{z}$ in terms of the reparameterization variable $\boldsymbol{\epsilon}$. We write $\mathsf{RepGrad}_\theta$ for the contribution from the first source, and $\mathsf{BouContr}_\theta$ for that from the second source. The proof of the theorem uses an existing but less known theorem about interchanging integration and differentiation under moving domain [3], together with the divergence theorem. It appears in the supplementary material of this paper.

At this point, some readers may feel uneasy with the $\mathsf{BouContr}_\theta$ term in our theorem. They may reason like this. Every boundary $\partial R_k$ is a measure-zero set in $\mathbb{R}^n$, and non-differentiabilities occur only at these $\partial R_k$'s. So, why do we need more than $\mathsf{RepGrad}_\theta$, the case-split version of the usual reparameterization? Unfortunately, this heuristic reasoning is incorrect, as indicated by the following proposition:

**Proposition 2.** *There are models satisfying this section's conditions s.t.* $\nabla_\theta \mathsf{ELBO}_\theta \neq \mathsf{RepGrad}_\theta$.

*Proof.* Consider the model $p(x,z) = \mathcal{N}(z|0,1)\big(\mathbb{1}[z>0]\mathcal{N}(x|5,1) + \mathbb{1}[z\leq 0]\mathcal{N}(x|-2,1)\big)$ for $x \in \mathbb{R}$ and $z \in \mathbb{R}$, the variational distribution $q_\theta(z) = \mathcal{N}(z|\theta,1)$ for $\theta \in \mathbb{R}$, and its reparameterization $f_\theta(\epsilon) = \epsilon + \theta$ and $q(\epsilon) = \mathcal{N}(\epsilon|0,1)$ for $\epsilon \in \mathbb{R}$. For an observed value $x^0 = 0$, the joint density $p(x^0, z)$ becomes $r(z) = \mathbb{1}[z>0] \cdot c_1 \mathcal{N}(z|0,1) + \mathbb{1}[z\leq 0] \cdot c_2 \mathcal{N}(z|0,1)$, where $c_1 = \mathcal{N}(0|5,1)$ and $c_2 = \mathcal{N}(0|-2,1)$. Notice that $r$ is non-differentiable only at $z = 0$ and $\{0\}$ is a null set in $\mathbb{R}$.

For any $\theta$, $\nabla_\theta \mathsf{ELBO}_\theta$ is computed as follows: Since $\log(r(z)/q_\theta(z)) = \mathbb{1}[z>0] \cdot (\theta^2/2 - z\theta + \log c_1) + \mathbb{1}[z\leq 0] \cdot (\theta^2/2 - z\theta + \log c_2)$, we have[1] $\mathsf{ELBO}_\theta = \frac{1}{2}[-\theta^2 + \mathrm{erf}(\theta/\sqrt{2})\log(c_1/c_2) + \log(c_1 c_2)]$ and thus obtain $\nabla_\theta \mathsf{ELBO}_\theta = -\theta + \log(c_1/c_2)\exp(-\theta^2/2)/\sqrt{2\pi}$.

On the other hand, $\mathsf{RepGrad}_\theta$ is computed as follows: After reparameterizing $z$ into $\epsilon$, we have $\log\big(r(f_\theta(\epsilon))/q_\theta(f_\theta(\epsilon))\big) = \mathbb{1}[\epsilon + \theta > 0]\cdot(-\theta^2/2 - \epsilon\theta + \log c_1) + \mathbb{1}[\epsilon + \theta \leq 0]\cdot(-\theta^2/2 - \epsilon\theta + \log c_2)$, so the term inside the expectation of $\mathsf{RepGrad}_\theta$ is $\mathbb{1}[\epsilon + \theta > 0] \cdot (-\theta - \epsilon) + \mathbb{1}[\epsilon + \theta \leq 0] \cdot (-\theta - \epsilon)$ and we obtain $\mathsf{RepGrad}_\theta = -\theta$.

Note that $\nabla_\theta \mathsf{ELBO}_\theta \neq \mathsf{RepGrad}_\theta$ for any $\theta$. The difference between the two quantities is $\mathsf{BouContr}_\theta$ in Theorem 1. The main culprit here is that interchanging differentiation and integration is sometimes invalid: for $D_1, D_2(\theta) \subset \mathbb{R}^n$ and $\alpha_1, \alpha_2 : \mathbb{R}^n \times \mathbb{R}^d \to \mathbb{R}$, the below equations *do not* hold in general if $\alpha_1$ is not differentiable in $\theta$, and if $D_2(\cdot)$ is not constant (even when $\alpha_2$ is differentiable in $\theta$).

$$\nabla_\theta \int_{D_1} \alpha_1(\boldsymbol{\epsilon}, \theta)d\boldsymbol{\epsilon} = \int_{D_1} \nabla_\theta \alpha_1(\boldsymbol{\epsilon}, \theta)d\boldsymbol{\epsilon} \quad \text{and} \quad \nabla_\theta \int_{D_2(\theta)} \alpha_2(\boldsymbol{\epsilon}, \theta)d\boldsymbol{\epsilon} = \int_{D_2(\theta)} \nabla_\theta \alpha_2(\boldsymbol{\epsilon}, \theta)d\boldsymbol{\epsilon}.$$

□

The $\mathsf{RepGrad}_\theta$ term in Theorem 1 can be easily estimated by the standard Monte Carlo:

$$\mathsf{RepGrad}_\theta \approx \frac{1}{N}\sum_{i=1}^N \left(\sum_{k=1}^K \mathbb{1}\big[f_\theta(\boldsymbol{\epsilon}^i) \in R_k\big] \cdot \nabla_\theta h_k(\boldsymbol{\epsilon}^i, \theta)\right) \quad \text{for i.i.d. } \boldsymbol{\epsilon}^1, \ldots, \boldsymbol{\epsilon}^N \sim q(\boldsymbol{\epsilon}).$$

We write $\widehat{\mathsf{RepGrad}_\theta}$ for this estimate.

However, estimating the other $\mathsf{BouContr}_\theta$ term is not that easy, because of the difficulties in estimating surface integrals in the term. In general, to approximate a surface integral well, we need a parameterization of the surface, and a scheme for generating samples from it [2]; this general methodology and a known theorem related to our case are reviewed in the supplementary material.

In this paper, we focus on a class of models that use relatively simple (reparameterized) boundaries $f_\theta^{-1}(\partial R_k)$ and permit, as a result, an efficient method for estimating surface integrals in $\mathsf{BouContr}_\theta$.

A good way to understand our simple-boundary condition is to start with something even simpler, namely the condition that $f_\theta^{-1}(\partial R_k)$ is an $(n-1)$-dimensional hyperplane $\{\boldsymbol{\epsilon} \mid \boldsymbol{a} \cdot \boldsymbol{\epsilon} = c\}$. Here the operation $\cdot$ denotes the dot-product. A surface integral over such a hyperplane can be estimated using the following theorem:

**Theorem 3.** *Let* $q(\boldsymbol{\epsilon}) = \prod_{i=1}^n q(\boldsymbol{\epsilon}_i)$ *and* $S$ *a measurable subset of* $\mathbb{R}^n$. *Assume that* $S = \{\boldsymbol{\epsilon} \mid \boldsymbol{a} \cdot \boldsymbol{\epsilon} > c\}$ *or* $S = \{\boldsymbol{\epsilon} \mid \boldsymbol{a} \cdot \boldsymbol{\epsilon} \geq c\}$ *for some* $\boldsymbol{a} \in \mathbb{R}^n$ *and* $c \in \mathbb{R}$, *and that* $\boldsymbol{a}_j \neq 0$ *for some* $j$. *Then,*

$$\int_{\partial S} \big(q(\boldsymbol{\epsilon})\boldsymbol{F}(\boldsymbol{\epsilon})\big) \bullet d\boldsymbol{\Sigma} = \mathbb{E}_{q(\boldsymbol{\zeta})}\left[\boldsymbol{G}(g(\boldsymbol{\zeta})) \bullet \boldsymbol{n}\right] \quad \text{for all measurable } \boldsymbol{F} : \mathbb{R}^n \to \mathbb{R}^{d \times n}.$$

*Here* $d\boldsymbol{\Sigma}$ *is the normal vector pointing outward with respect to* $S$, $\boldsymbol{\zeta}$ *ranges over* $\mathbb{R}^{n-1}$, *its density* $q(\boldsymbol{\zeta})$ *is the product of the densities for its components, and this component density* $q(\boldsymbol{\zeta}_i)$ *is the same as the density* $q(\boldsymbol{\epsilon}_{i'})$ *for the* $i'$*-th component of* $\boldsymbol{\epsilon}$, *where* $i' = i + \mathbb{1}[i \geq j]$. *Also,*

$$\boldsymbol{G}(\boldsymbol{\epsilon}) \triangleq q(\boldsymbol{\epsilon}_j)\boldsymbol{F}(\boldsymbol{\epsilon}), \qquad g(\boldsymbol{\zeta}) \triangleq \left(\boldsymbol{\zeta}_1, \ldots, \boldsymbol{\zeta}_{j-1}, \frac{1}{\boldsymbol{a}_j}(c - \boldsymbol{a}_{-j} \cdot \boldsymbol{\zeta}), \boldsymbol{\zeta}_j, \ldots, \boldsymbol{\zeta}_{n-1}\right)^\mathsf{T},$$

$$\boldsymbol{a}_{-j} \triangleq (\boldsymbol{a}_1, \ldots, \boldsymbol{a}_{j-1}, \boldsymbol{a}_{j+1}, \ldots, \boldsymbol{a}_n), \qquad \boldsymbol{n} \triangleq \mathrm{sgn}(-\boldsymbol{a}_j)\left(\frac{\boldsymbol{a}_1}{\boldsymbol{a}_j}, \ldots, \frac{\boldsymbol{a}_{j-1}}{\boldsymbol{a}_j}, 1, \frac{\boldsymbol{a}_{j+1}}{\boldsymbol{a}_j}, \ldots, \frac{\boldsymbol{a}_n}{\boldsymbol{a}_j}\right)^\mathsf{T}.$$

The theorem says that if the boundary $\partial S$ is an $(n-1)$-dimensional hyperplane $\{\boldsymbol{\epsilon} \mid \boldsymbol{a} \cdot \boldsymbol{\epsilon} = c\}$, we can parameterize the surface by a linear map $g : \mathbb{R}^{n-1} \to \mathbb{R}^n$ and express the surface integral as an expectation over $q(\boldsymbol{\zeta})$. This distribution for $\boldsymbol{\zeta}$ is the marginalization of $q(\boldsymbol{\epsilon})$ over the $j$-th component. Inside the expectation, we have the product of the matrix $\boldsymbol{G}$ and the vector $\boldsymbol{n}$. The matrix comes from the integrand of the surface integral, and the vector is the direction of the surface. Note that $\boldsymbol{G}(\boldsymbol{\epsilon})$ has $q(\boldsymbol{\epsilon}_j)$ instead of $q(\boldsymbol{\epsilon})$; the missing part of $q(\boldsymbol{\epsilon})$ has been converted to the distribution $q(\boldsymbol{\zeta})$.

When every $f_\theta^{-1}(\partial R_k)$ is an $(n-1)$-dimensional hyperplane $\{\boldsymbol{\epsilon} \mid \boldsymbol{a} \cdot \boldsymbol{\epsilon} = c\}$ for $\boldsymbol{a} \in \mathbb{R}^n$ and $c \in \mathbb{R}$ with $\boldsymbol{a}_{j_k} \neq 0$, we can use Theorem 3 and estimate the surface integrals in $\mathsf{BouContr}_\theta$ as follows:

$$\int_{f_\theta^{-1}(\partial R_k)} \left( q(\boldsymbol{\epsilon}) h_k(\boldsymbol{\epsilon}, \theta) \boldsymbol{V}(\boldsymbol{\epsilon}, \theta) \right) \bullet d\boldsymbol{\Sigma} \approx \frac{1}{M} \sum_{i=1}^{M} \boldsymbol{W}(g(\boldsymbol{\zeta}^i)) \bullet \boldsymbol{n} \quad \text{for i.i.d. } \boldsymbol{\zeta}^1, \ldots, \boldsymbol{\zeta}^M \sim q(\boldsymbol{\zeta}),$$

where $\boldsymbol{W}(\boldsymbol{\epsilon}) = q(\boldsymbol{\epsilon}_{j_k}) h_k(\boldsymbol{\epsilon}, \theta) \boldsymbol{V}(\boldsymbol{\epsilon}, \theta)$. Let $\widehat{\mathsf{BouContr}_{(\theta,k)}}$ be this estimate. Then, our estimator for the gradient of $\mathsf{ELBO}_\theta$ in this case computes:

$$\widehat{\nabla_\theta \mathsf{ELBO}_\theta} \triangleq \widehat{\mathsf{RepGrad}_\theta} + \sum_{k=1}^{K} \widehat{\mathsf{BouContr}_{(\theta,k)}}.$$

The estimator is unbiased because of Theorems 1 and 3:

**Corollary 4.** $\mathbb{E}\left[ \widehat{\nabla_\theta \mathsf{ELBO}_\theta} \right] = \nabla_\theta \mathsf{ELBO}_\theta.$

We now relax the condition that each boundary is a hyperplane, and consider a more liberal *simple-boundary condition*, which is often satisfied by non-differentiable models from a first-order loop-free probabilistic programming language. This new condition and the estimator under this condition are what we have used in our implementation. The relaxed condition is that the regions $\{f_\theta^{-1}(R_k)\}_{1 \leq k \leq K}$ are obtained by partitioning $\mathbb{R}^n$ with $L$ $(n-1)$-dimensional hyperplanes. That is, there are affine maps $\Phi_1, \ldots, \Phi_L : \mathbb{R}^n \to \mathbb{R}$ such that for all $1 \leq k \leq K$,

$$f_\theta^{-1}(R_k) = \bigcap_{l=1}^{L} S_{l,(\sigma_k)_l} \qquad \text{for some } \sigma_k \in \{-1, 1\}^L$$

where $S_{l,1} = \{\boldsymbol{\epsilon} \mid \Phi_l(\boldsymbol{\epsilon}) > 0\}$ and $S_{l,-1} = \{\boldsymbol{\epsilon} \mid \Phi_l(\boldsymbol{\epsilon}) \leq 0\}$. Each affine map $\Phi_l$ defines an $(n-1)$-dimensional hyperplane $\partial S_{l,1}$, and $(\sigma_k)_l$ specifies on which side the region $f_\theta^{-1}(R_k)$ lies with respect to the hyperplane $\partial S_{l,1}$. Every probabilistic model written in a first-order probabilistic programming language satisfies the relaxed condition, if the model does not contain a loop and uses only a fixed finite number of random variables and the branch condition of each if statement in the model is linear in the latent variable $\boldsymbol{z}$; in such a case, $L$ is the number of if statements in the model.

Under the new condition, how can we estimate $\mathsf{BouContr}_\theta$? A naive approach is to estimate the $k$-th surface integral for each $k$ (in some way) and sum them up. However, with $L$ hyperplanes, the number $K$ of regions can grow as fast as $\mathcal{O}\left(L^n\right)$, implying that the naive approach is slow. Even worse the boundaries $f_\theta^{-1}(\partial R_k)$ do not satisfy the condition of Theorem 3, and just estimating the surface integral over each $f_\theta^{-1}(\partial R_k)$ may be difficult.

A solution is to transform the original formulation of $\mathsf{BouContr}_\theta$ such that it can be expressed as the sum of surface integrals over $\partial S_{l,1}$'s. The transformation is based on the following derivation:

$$\mathsf{BouContr}_\theta = \sum_{k=1}^{K} \int_{f_\theta^{-1}(\partial R_k)} \left( q(\boldsymbol{\epsilon}) h_k(\boldsymbol{\epsilon}, \theta) \boldsymbol{V}(\boldsymbol{\epsilon}, \theta) \right) \bullet d\boldsymbol{\Sigma}$$

$$= \sum_{l=1}^{L} \int_{\partial S_{l,1}} \left( q(\boldsymbol{\epsilon}) \boldsymbol{V}(\boldsymbol{\epsilon}, \theta) \sum_{k=1}^{K} \mathbb{1}\left[\boldsymbol{\epsilon} \in \overline{f_\theta^{-1}(R_k)}\right] (\sigma_k)_l h_k(\boldsymbol{\epsilon}, \theta) \right) \bullet d\boldsymbol{\Sigma} \qquad (4)$$

where $\overline{T}$ denotes the closure of $T \subset \mathbb{R}^n$, and $d\boldsymbol{\Sigma}$ in (4) is the normal vector pointing outward with respect to $S_{l,1}$. Since $\{f_\theta^{-1}(R_k)\}_k$ are obtained by partitioning $\mathbb{R}^n$ with $\{\partial S_{l,1}\}_l$, we can rearrange the sum of $K$ surface integrals over complicated boundaries $f_\theta^{-1}(\partial R_k)$, into the sum of $L$ surface integrals over the hyperplanes $\partial S_{l,1}$ as above. Although the expression inside the summation over $k$ in (4) looks complicated, for almost all $\boldsymbol{\epsilon}$, the indicator function is nonzero for exactly two $k$'s: $k_1$

with $(\sigma_{k_1})_l = 1$ and $k_{-1}$ with $(\sigma_{k_{-1}})_l = -1$. So, we can efficiently estimate the $l$-th surface integral in (4) using Theorem 3, and call this estimate $\widehat{\text{BouContr}_{(\theta,l)}}'$. Then, our estimator for the gradient of $\text{ELBO}_\theta$ in this more general case computes:

$$\widehat{\nabla_\theta\text{ELBO}_\theta}' \triangleq \widehat{\text{RepGrad}_\theta} + \sum_{l=1}^{L} \widehat{\text{BouContr}_{(\theta,l)}}'. \tag{5}$$

The estimator is unbiased because of Theorems 1 and 3 and Equation 4:

**Corollary 5.** $\mathbb{E}\left[\widehat{\nabla_\theta\text{ELBO}_\theta}'\right] = \nabla_\theta\text{ELBO}_\theta.$

## 4 Experimental Evaluation

We experimentally compare our gradient estimator (OURS) to the score estimator (SCORE), an unbiased gradient estimator that is applicable to non-differentiable models, and the reparameterization estimator (REPARAM), a biased gradient estimator that computes only $\widehat{\text{RepGrad}_\theta}$ (discussed in Section 3). REPARAM is biased in our experiments because it is applied to non-differentiable models.

We implemented a black-box variational inference engine that accepts a probabilistic model written in a simple probabilistic programming language (which supports basic constructs such as `sample`, `observe`, and `if` statements) and performs variational inference using one of the three aforementioned gradient estimators. Our implementation[2] is written in Python and uses `autograd` [18], an automatic differentiation package for Python, to automatically compute the gradient term in $\widehat{\text{RepGrad}_\theta}$ for an arbitrary probabilistic model.

**Benchmarks.** We evaluate our estimator on three models for small sequential data:

- `temperature` [33] models the random dynamics of a controller that attempts to keep the temperature of a room within specified bounds. The controller's state has a continuous part for the room temperature and a discrete part that records the on or off of an air conditioner. At each time step, the value of this discrete part decides which of two different random state updates is employed, and incurs the non-differentiability of the model's density. We use a synthetically-generated sequence of 21 noisy measurements of temperatures, and perform posterior inference on the sequence of the controller's states given these noisy measurements. This model consists of a 41-dimensional latent variable and 80 if statements.
- `textmsg` [1] is a model for the numbers of per-day SNS messages over the period of 74 days (skipping every other day). It allows the SNS-usage pattern to change over the period, and this change causes non-differentiability. Finding the posterior distribution over this change is the goal of the inference problem in this case. We use the data from [1]. This model consists of a 3-dimensional latent variable and 37 if statements.
- `influenza` [32] is a model for the US influenza mortality data in 1969. The mortality rate in each month depends on whether the dominant influenza virus is of type 1 or 2, and finding this type information from a sequence of observed mortality rates is the goal of the inference. The virus type is the cause of non-differentiability in this example. This model consists of a 37-dimensional latent variable and 24 if statements.

**Experimental setup.** We optimize the ELBO objective using Adam [11] with two stepsizes: 0.001 and 0.01. We run Adam for 10000 iterations and at each iteration, we compute each estimator using $N \in \{1, 8, 16\}$ Monte Carlo samples. For OURS, we use a single subsample $l$ (drawn uniformly at random from $\{1, \cdots, L\}$) to estimate the summation in (5), and use $N$ Monte Carlo samples to compute $\widehat{\text{BouContr}_{(\theta,l)}}'$. While maximizing ELBO, we measure two things: the variance of estimated gradients of ELBO, and ELBO itself. Since each gradient is not scalar, we measure two kinds of variance of the gradient, as in [23]: $\text{Avg}(\mathbb{V}(\cdot))$, the average variance of each of its components, and $\mathbb{V}(\|\cdot\|_2)$, the variance of its $l^2$-norm. To estimate the variances and the ELBO objective, we use 16 and 1000 Monte Carlo samples, respectively.

| Estimator | Type of Variance | temperature | textmsg | influenza |
|---|---|---|---|---|
| REPARAM | $\mathrm{Avg}(\mathbb{V}(\cdot))$ | $\mathbf{4.45 \times 10^{-9}}$ | $2.91 \times 10^{-2}$ | $\mathbf{4.38 \times 10^{-3}}$ |
| | $\mathbb{V}(\|\cdot\|_2)$ | $\mathbf{2.45 \times 10^{-8}}$ | $2.92 \times 10^{-2}$ | $\mathbf{2.12 \times 10^{-3}}$ |
| OURS | $\mathrm{Avg}(\mathbb{V}(\cdot))$ | $1.85 \times 10^{-6}$ | $\mathbf{2.77 \times 10^{-2}}$ | $4.89 \times 10^{-3}$ |
| | $\mathbb{V}(\|\cdot\|_2)$ | $7.59 \times 10^{-5}$ | $\mathbf{2.46 \times 10^{-2}}$ | $2.36 \times 10^{-3}$ |

(a) stepsize $= 0.001$

| Estimator | Type of Variance | temperature | textmsg | influenza |
|---|---|---|---|---|
| REPARAM | $\mathrm{Avg}(\mathbb{V}(\cdot))$ | $3.88 \times 10^{-11}$ | $\mathbf{5.03 \times 10^{-4}}$ | $\mathbf{2.46 \times 10^{-3}}$ |
| | $\mathbb{V}(\|\cdot\|_2)$ | $\mathbf{6.11 \times 10^{-11}}$ | $1.02 \times 10^{-3}$ | $\mathbf{1.26 \times 10^{-3}}$ |
| OURS | $\mathrm{Avg}(\mathbb{V}(\cdot))$ | $\mathbf{1.24 \times 10^{-11}}$ | $5.07 \times 10^{-4}$ | $2.80 \times 10^{-3}$ |
| | $\mathbb{V}(\|\cdot\|_2)$ | $8.05 \times 10^{-11}$ | $\mathbf{8.12 \times 10^{-4}}$ | $1.40 \times 10^{-3}$ |

(b) stepsize $= 0.01$

Table 1: Ratio of {REPARAM, OURS}'s average variance to SCORE's for $N = 1$. The values for SCORE are all 1, so omitted. The optimization trajectories used to compute the above variances are shown in Figure 1.

| Estimator | temperature | textmsg | influenza |
|---|---|---|---|
| SCORE | 21.7 | 4.9 | 18.7 |
| REPARAM | 46.1 | 15.4 | 251.4 |
| OURS | 79.2 | 24.9 | 269.8 |

Table 2: Computation time (in ms) per iteration for $N = 1$.

**Results.** Table 1 compares the average variance of each estimator for $N = 1$, where the average is taken over a single optimization trajectory. The table clearly shows that during the optimization process, OURS has several orders of magnitude (sometimes $< 10^{-10}$ times) smaller variances than SCORE. Since OURS computes additional terms when compared with REPARAM, we expect that OURS would have larger variances than REPARAM, and this is confirmed by the table. It is noteworthy, however, that for most benchmarks, the averaged variances of OURS are very close to those of REPARAM. This suggests that the additional term $\mathrm{BouContr}_\theta$ in our estimator often introduces much smaller variances than the reparameterization term $\mathrm{RepGrad}_\theta$.

Figure 1 shows the ELBO objective, for different estimators with different $N$'s, as a function of the iteration number. As expected, using a larger $N$ makes all estimators converge faster in a more stable manner. In all three benchmarks, OURS outperforms (or performs similarly to) the other two and converges stably, and REPARAM beats SCORE. Increasing the stepsize to $0.01$ makes SCORE unstable in temperature and textmsg. It is also worth noting that REPARAM converges to sub-optimal values in temperature (possibly because REPARAM is biased).

Table 2 shows the computation time per iteration of each approach for $N = 1$. Our implementation performs the worst in this wall-time comparison, but the gap between OURS and REPARAM is not huge: the computation time of OURS is less than $1.72$ times that of REPARAM in all benchmarks. Furthermore, we want to point out that our implementation is an early unoptimized prototype, and there are several rooms to improve in the implementation. For instance, it currently constructs Python functions dynamically, and computes the gradients of these functions using autograd. But this dynamic approach is costly because autograd is not optimized for such dynamically constructed functions; this can also be observed in the bad performance of REPARAM, particularly in influenza, that employs the same strategy of dynamically constructing functions and taking their gradients. So one possible optimization is to avoid this gradient computation of dynamically constructed functions by building the functions statically during compilation.

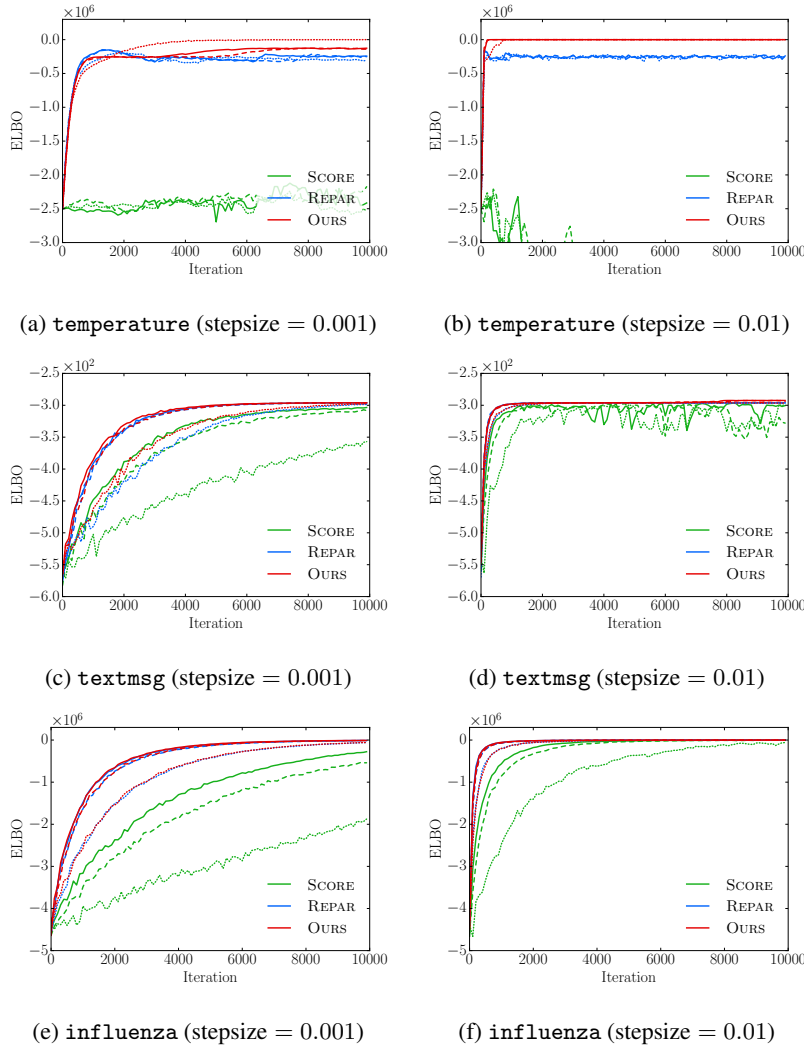

Figure 1: The ELBO objective as a function of the iteration number. {dotted, dashed, solid} lines represent $\{N = 1, N = 8, N = 16\}$.

## 5 Related Work

A common example of a model with a non-differentiable density is the one that uses discrete random variables, typically together with continuous random variables.[3] Coming up with an efficient algorithm for stochastic variational inference for such a model has been an active research topic. Maddison *et al.* [20] and Jang *et al.* [10] proposed continuous relaxations of discrete random variables that convert non-differentiable variational objectives to differentiable ones and make the reparameterization trick applicable. Also, a variety of control variates for the standard score estimator [35, 27, 36, 28] for the gradients of variational objectives have been developed [28, 7, 8, 34, 6, 23], some of which use biased yet differentiable control variates such that the reparameterization trick can be used to correct the bias [7, 34, 6].

Our work extends this line of research by adding a version of the reparameterization trick that can be applied to models with discrete random variables. For instance, consider a model $p(x, z)$ with $z$ discrete. By applying the Gumbel-Max reparameterization [9, 21] to $z$, we transform $p(x, z)$ to $p(x, z, c)$, where $c$ is sampled from the Gumbel distribution and $z$ in $p(x, z, c)$ is defined determin-

istically from $c$ using the $\arg\max$ operation. Since $\arg\max$ can be written as if statements, we can express $p(x, z, c)$ in the form of (3) to which our reparameterization gradient can be applied. Investigating the effectiveness of this approach for discrete random variables is an interesting topic for future research.

The reparameterization trick was initially used with normal distribution [13, 30], but its scope was soon extended to other common distributions, such as gamma, Dirichlet, and beta [14, 31, 26]. Techniques for constructing normalizing flow [29, 12] can also be viewed as methods for creating distributions in a reparameterized form. In the paper, we did not consider these recent developments and mainly focused on the reparameterization with normal distribution. One interesting future avenue is to further develop our approach for these other reparameterization cases. We expect that the main challenge will be to find an effective method for handling the surface integrals in Theorem 1.

## 6 Conclusion

We have presented a new estimator for the gradient of the standard variational objective, ELBO. The key feature of our estimator is that it can keep variance under control by using a form of the reparameterization trick even when the density of a model is not differentiable. The estimator splits the space of the latent random variable into a lower-dimensional subspace where the density may fail to be differentiable, and the rest where the density is differentiable. Then, it estimates the contributions of both parts to the gradient separately, using a version of manifold sampling for the former and the reparameterization trick for the latter. We have shown the unbiasedness of our estimator using a theorem for interchanging integration and differentiation under moving domain [3] and the divergence theorem. Also, we have experimentally demonstrated the promise of our estimator using three time-series models. One interesting future direction is to investigate the possibility of applying our ideas to recent variational objectives [24, 17, 19, 16, 25], which are based on tighter lower bounds of marginal likelihood than the standard ELBO.

When viewed from a high level, our work suggests a heuristic of splitting the latent space into a bad yet tiny subspace and the remaining good one, and solving an estimation problem in each subspace separately. The latter subspace has several good properties and so it may allow the use of efficient estimation techniques that exploit those properties. The former subspace is, on the other hand, tiny and the estimation error from the subspace may, therefore, be relatively small. We would like to explore this heuristic and its extension in different contexts, such as stochastic variational inference with different objectives [24, 17, 19, 16, 25].

**Acknowledgments**

We thank Hyunjik Kim, George Tucker, Frank Wood and anonymous reviewers for their helpful comments, and Shin Yoo and Seongmin Lee for allowing and helping us to use their cluster machines. This research was supported by the Engineering Research Center Program through the National Research Foundation of Korea (NRF) funded by the Korean Government MSIT (NRF-2018R1A5A1059921), and also by Next-Generation Information Computing Development Program through the National Research Foundation of Korea (NRF) funded by the Ministry of Science, ICT (2017M3C4A7068177).

## Footnotes

[1] The error function erf is defined by $\mathrm{erf}(x) = 2\int_0^x \exp(-t^2)\,dt/\sqrt{\pi}$.

[2] Code is available at `https://github.com/wonyeol/reparam-nondiff`.

[3] Another common example of such a model is the one that uses if statements whose branch conditions contain continuous random variables, which is the main focus of our work.

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
