[Supplementary Material]

# Supplementary Material: Reparameterization Gradient for Non-differentiable Models

## A Proof of Theorem 1

Using reparameterization, we can write $\mathsf{ELBO}_\theta$ as follows:

$$
\begin{aligned}
\mathsf{ELBO}_\theta &= \mathbb{E}_{q(\boldsymbol{\epsilon})}\left[\log \frac{\sum_{k=1}^K \mathbb{1}[f_\theta(\boldsymbol{\epsilon}) \in R_k] \cdot r_k(f_\theta(\boldsymbol{\epsilon}))}{q_\theta(f_\theta(\boldsymbol{\epsilon}))}\right] \\
&= \mathbb{E}_{q(\boldsymbol{\epsilon})}\left[\sum_{k=1}^K \mathbb{1}[f_\theta(\boldsymbol{\epsilon}) \in R_k] \cdot \log \frac{r_k(f_\theta(\boldsymbol{\epsilon}))}{q_\theta(f_\theta(\boldsymbol{\epsilon}))}\right] \\
&= \sum_{k=1}^K \mathbb{E}_{q(\boldsymbol{\epsilon})}\left[\mathbb{1}[f_\theta(\boldsymbol{\epsilon}) \in R_k] \cdot h_k(\boldsymbol{\epsilon}, \theta)\right].
\end{aligned}
\tag{6}
$$

In (6), we can move the summation and the indicator function out of $\log$ since the regions $\{R_k\}_{1 \le k \le K}$ are disjoint. We then compute the gradient of $\mathsf{ELBO}_\theta$ as follows:

$$
\begin{aligned}
&\nabla_\theta \mathsf{ELBO}_\theta \\
&= \sum_{k=1}^K \nabla_\theta \mathbb{E}_{q(\boldsymbol{\epsilon})}\left[\mathbb{1}[f_\theta(\boldsymbol{\epsilon}) \in R_k] \cdot h_k(\boldsymbol{\epsilon}, \theta)\right] \\
&= \sum_{k=1}^K \nabla_\theta \int_{f_\theta^{-1}(R_k)} q(\boldsymbol{\epsilon}) h_k(\boldsymbol{\epsilon}, \theta) d\boldsymbol{\epsilon} \\
&= \sum_{k=1}^K \int_{f_\theta^{-1}(R_k)} \Big(q(\boldsymbol{\epsilon})\nabla_\theta h_k(\boldsymbol{\epsilon}, \theta) + \nabla_{\boldsymbol{\epsilon}} \bullet \big(q(\boldsymbol{\epsilon}) h_k(\boldsymbol{\epsilon}, \theta)\boldsymbol{V}(\boldsymbol{\epsilon}, \theta)\big)\Big)d\boldsymbol{\epsilon} \\
&= \mathbb{E}_{q(\boldsymbol{\epsilon})}\left[\sum_{k=1}^K \mathbb{1}[f_\theta(\boldsymbol{\epsilon}) \in R_k] \cdot \nabla_\theta h_k(\boldsymbol{\epsilon}, \theta)\right] + \sum_{k=1}^K \int_{f_\theta^{-1}(R_k)} \nabla_{\boldsymbol{\epsilon}} \bullet \big(q(\boldsymbol{\epsilon}) h_k(\boldsymbol{\epsilon}, \theta)\boldsymbol{V}(\boldsymbol{\epsilon}, \theta)\big)d\boldsymbol{\epsilon} \\
&= \underbrace{\mathbb{E}_{q(\boldsymbol{\epsilon})}\left[\sum_{k=1}^K \mathbb{1}[f_\theta(\boldsymbol{\epsilon}) \in R_k] \cdot \nabla_\theta h_k(\boldsymbol{\epsilon}, \theta)\right]}_{\mathsf{RepGrad}_\theta} + \underbrace{\sum_{k=1}^K \int_{f_\theta^{-1}(\partial R_k)} \big(q(\boldsymbol{\epsilon}) h_k(\boldsymbol{\epsilon}, \theta)\boldsymbol{V}(\boldsymbol{\epsilon}, \theta)\big) \bullet d\boldsymbol{\Sigma}}_{\mathsf{BouContr}_\theta}
\end{aligned}
\tag{7}
\tag{8}
$$

where $\nabla_{\boldsymbol{\epsilon}} \bullet \boldsymbol{U}$ denotes the column vector whose $i$-th component is $\nabla_{\boldsymbol{\epsilon}} \cdot \boldsymbol{U}_i$, the divergence of $\boldsymbol{U}_i$ with respect to $\boldsymbol{\epsilon}$. (8) is the formula that we wanted to prove.

The two non-trivial steps in the above derivation are (7) and (8). First, (7) is a direct consequence of the following theorem, existing yet less well-known, on exchanging integration and differentiation under moving domain:

**Theorem 6.** *Let $D_\theta \subset \mathbb{R}^n$ be a smoothly parameterized region. That is, there exist open sets $\Omega \subset \mathbb{R}^n$ and $\Theta \subset \mathbb{R}$, and twice continuously differentiable $\widehat{\boldsymbol{\epsilon}} : \Omega \times \Theta \to \mathbb{R}^n$ such that $D_\theta = \widehat{\boldsymbol{\epsilon}}(\Omega, \theta)$ for each $\theta \in \Theta$. Suppose that $\widehat{\boldsymbol{\epsilon}}(\cdot, \theta)$ is a $C^1$-diffeomorphism for each $\theta \in \Theta$. Let $f : \mathbb{R}^n \times \mathbb{R} \to \mathbb{R}$ be a differentiable function such that $f(\cdot, \theta) \in \mathcal{L}^1(D_\theta)$ for each $\theta \in \Theta$. If there exists $g : \Omega \to \mathbb{R}$ such that $g \in \mathcal{L}^1(\Omega)$ and $\left|\nabla_\theta\big(f(\widehat{\boldsymbol{\epsilon}}, \theta)\big|\frac{\partial \widehat{\boldsymbol{\epsilon}}}{\partial \boldsymbol{\omega}}\big|\big)\right| \le g(\boldsymbol{\omega})$ for any $\theta \in \Theta$ and $\boldsymbol{\omega} \in \Omega$, then*

$$
\nabla_\theta \int_{D_\theta} f(\boldsymbol{\epsilon}, \theta) d\boldsymbol{\epsilon} = \int_{D_\theta} \Big(\nabla_\theta f + \nabla_{\boldsymbol{\epsilon}} \cdot (f\mathbf{v})\Big)(\boldsymbol{\epsilon}, \theta) d\boldsymbol{\epsilon}.
$$

*Here $\mathbf{v}(\boldsymbol{\epsilon}, \theta)$ denotes $\nabla_\theta \widehat{\boldsymbol{\epsilon}}(\boldsymbol{\omega}, \theta)\big|_{\boldsymbol{\omega} = \widehat{\boldsymbol{\epsilon}}_\theta^{-1}(\boldsymbol{\epsilon})}$, the velocity of the particle $\boldsymbol{\epsilon}$ at time $\theta$.*

The statement of Theorem 6 (without detailed conditions as we present above) and the sketch of its proof can be found in [3]. One subtlety in applying Theorem 6 to our case is that $R_k$ (which corresponds to $\Omega$ in the theorem) may not be open, so the theorem may not be immediately applicable. However, since the boundary $\partial R_k$ has Lebesgue measure zero in $\mathbb{R}^n$, ignoring the reparameterized boundary $f_\theta^{-1}(\partial R_k)$ in the integral of (7) does not change the value of the integral. Hence, we apply Theorem 6 to $D_\theta = \text{int}(f_\theta^{-1}(R_k))$ (which is possible because $\Omega = \text{int}(R_k)$ is now open), and this gives us the desired result. Here $\text{int}(T)$ denotes the interior of $T$.

Second, to prove (8), it suffices to show that

$$\int_V \nabla_\epsilon \bullet U(\epsilon) d\epsilon = \int_{\partial V} U(\epsilon) \bullet d\Sigma$$

where $U(\epsilon) = q(\epsilon) h_k(\epsilon, \theta) V(\epsilon, \theta)$ and $V = f_\theta^{-1}(R_k)$. To prove this equality, we apply the divergence theorem:

**Theorem 7** (Divergence theorem). *Let $V$ be a compact subset of $\mathbb{R}^n$ that has a piecewise smooth boundary $\partial V$. If $F$ is a differentiable vector field defined on a neighborhood of $V$, then*

$$\int_V (\nabla \cdot F) \, dV = \int_{\partial V} F \cdot d\Sigma$$

*where $d\Sigma$ is the outward pointing normal vector of the boundary $\partial V$.*

In our case, the region $V = f_\theta^{-1}(R_k)$ may not be compact, so we cannot directly apply Theorem 7 to $U$. To circumvent the non-compactness issue, we assume that $q(\epsilon)$ is in $\mathcal{S}(\mathbb{R}^n)$, the Schwartz space on $\mathbb{R}^n$. That is, assume that every partial derivative of $q(\epsilon)$ of any order decays faster than any polynomial. This assumption is reasonable in that the probability density of many important probability distributions (e.g., the normal distribution) is in $\mathcal{S}(\mathbb{R}^n)$. Since $q \in \mathcal{S}(\mathbb{R}^n)$, there exists a sequence of test functions $\{\phi_j\}_{j \in \mathbb{N}}$ such that each $\phi_j$ has compact support and $\{\phi_j\}_{j \in \mathbb{N}}$ converges to $q$ in $\mathcal{S}(\mathbb{R}^n)$, which is a well-known result in functional analysis. Since each $\phi_j$ has compact support, so does $U^j(\epsilon) \triangleq \phi_j(\epsilon) h_k(\epsilon, \theta) V(\epsilon, \theta)$. By applying Theorem 7 to $U^j$, we have

$$\int_V \nabla_\epsilon \bullet U^j(\epsilon) d\epsilon = \int_{\partial V} U^j(\epsilon) \bullet d\Sigma.$$

Because $\{\phi_j\}_{j \in \mathbb{N}}$ converges to $q$ in $\mathcal{S}(\mathbb{R}^n)$, taking the limit $j \to \infty$ on the both sides of the equation gives us the desired result.

# B    Proof of Theorem 3

Theorem 3 is a direct consequence of the following theorem called "area formula":

**Theorem 8** (Area formula). *Suppose that $g : \mathbb{R}^{n-1} \to \mathbb{R}^n$ is injective and Lipschitz. If $A \subset \mathbb{R}^{n-1}$ is measurable and $H : \mathbb{R}^n \to \mathbb{R}^n$ is measurable, then*

$$\int_{g(A)} H(\epsilon) \cdot d\Sigma = \int_A \Big(H(g(\zeta)) \cdot n(\zeta)\Big) |Jg(\zeta)| \, d\zeta$$

*where $Jg(\zeta) = \det\left[\frac{\partial g(\zeta)}{\partial \zeta_1} \big| \frac{\partial g(\zeta)}{\partial \zeta_2} \big| \cdots \big| \frac{\partial g(\zeta)}{\partial \zeta_{n-1}} \big| n(\zeta)\right]$, and $n(\zeta)$ is the unit normal vector of the hypersurface $g(A)$ at $g(\zeta)$ such that it has the same direction as $d\Sigma$.*

A more general version of Theorem 8 can be found in [2]. In our case, the hypersurface $g(A)$ for the surface integral on the LHS is given by $\{\epsilon \mid a \cdot \epsilon = c\}$, so we use $A = \mathbb{R}^{n-1}$ and $g(\zeta) = \left(\zeta_1, \ldots, \zeta_{j-1}, \frac{1}{a_j}(c - a_{-j} \cdot \zeta), \zeta_j, \ldots, \zeta_{n-1}\right)^\mathsf{T}$ and apply Theorem 8 with $H(\epsilon) = q(\epsilon) F(\epsilon)$. In this settings, $n(\zeta)$ and $|Jg(\zeta)|$ are calculated as

$$n(\zeta) = \text{sgn}(-a_j) \frac{|a_j|}{\|a\|_2} \left(\frac{a_1}{a_j}, \ldots, \frac{a_{j-1}}{a_j}, 1, \frac{a_{j+1}}{a_j}, \ldots, \frac{a_n}{a_j}\right)^\mathsf{T} \quad \text{and} \quad |Jg(\zeta)| = \frac{\|a\|_2}{|a_j|},$$

and this gives us the desired result.