[Reviews · NeurIPS 2018]

Reviewer 1



The authors provide a new method for computing reparameterization gradients when the target distribution contains a finite number of non-differentiabilities. They derive unbiased gradient estimators for a class of simple non-differentiable densities, and show in experiments that these can lead to improved convergence results in the number of iterations. The idea is interesting, novel, and formalizes issues with interchanging integration and differentiation that might not be evident to most of the NIPS community. The paper is well-written and was easy to follow, even though it relies heavily on the supplementary material to formalize most proofs. It is still somewhat unclear to me whether the method can be made to be competitive also in wall-clock time. Still I believe the paper presents a method, derivation, and results that are illuminating and useful for the community. Minor Comments: - You define matrix product notation but not dot-product. Would be extra clear on e.g. line 152 to also define this notation because above you use dot to also represent scalar scalar multiplication. - Line 158, is this meant to be just i rather than i'? Or is there another definition of i'?

Reviewer 2



EDITED AFTER REBUTTAL PERIOD: The rebuttal addresses concerns 3 and 4. I like the plots in Figure 1' and suggest to include them in the paper, together with table 2'. Regarding concerns 1 and 2, I disagree with the authors' rebuttal in that Gumble-softmax (and related) approaches are "orthogonal" to the paper. As noted in the rebuttal, any model with categorical variables can be rewritten in terms of if/else statements. That is exactly what I had in mind when I wrote my initial review, which is why I suggested to include an experiment involving a model with categorical latent variables. ============================== In this paper, a VI stochastic gradient estimator for non-differentiable models is proposed. The proposed estimator contains two terms; one resembles the reparameterization gradient and the other involves surface integrals over the non-differentiable boundary. The estimator is compared on three models involving if statements to the score function estimator as well as a naive approach that takes into account the reparameterization component only. To the best of my knowledge, the method presented in this paper is novel. The paper is very well written and the ideas are easy to follow, although I am not familiar with the results in [3] about exchanging integration and differentiation under moving domain, so I can't really assess the correctness of the math and the proofs. They are technically sound, though. I have some comments and questions, mostly about the experimental evaluation. 1) I would have liked to see an experiment with categorical latent variables on a more "classical" model, similar to the ones considered in [9,32,6]. 2) I was missing a comparison with other gradient estimators for non-reparameterizable distributions, such as REBAR [32] and RELAX [6]. Can you say something about how these methods would compare? 3) I was confused about the numbers in Table 2. What does it mean exactly "to complete a single optimization trajectory"? I suggest to replace these numbers with the time per iteration of each approach, because otherwise it is hard to read if the numbers presented involve different number of iterations in each cell of the table. 4) I appreciate the honesty about the computation time being the worst for the proposed approach. However I would have liked to see a better implementation rather than an early unoptimized prototype. Based on the behavior of the "Repar" method in Figure 1, my guess is that the suggested idea of subsampling the L terms of the boundary component will perform well enough. I also suggest to implement at least one simple model in a library other than Python+autograd that avoids dynamic computational graphs. Even if the resulting implementation is not generic, it will allow to learn something about the true computational cost of each approach.

Reviewer 3



- The work seems interesting but it is at a preliminary stage. Credit to the authors for admitting this and for their honesty. - The method is sound and potentially useful (after having the run-time issue resolved). - As mentioned in the experiments and as was clear from the beginning of the methodology, the computational run-time is a massively impeding problem. - L153: "A surface integral over such a hyperplane can be estimated easily using the following theorem ...": Estimating the surface integral is still tricky and a major bottleneck. I do not think it can be "easily" estimated. - In equation (1), I reckon it is rather confusing and non-standard to keep the superscript 0 on x noting a specific example of x. - Writing of the paper has a considerable room for improvement. There are several typos and also the flow of ideas can improve. - Results of the experimental section are thoroughly analysed.